# Superhydrophobic and Thermally Conductive Coating for Restraining Corona Loss and Audible Noise of High-Voltage Transmission Lines

Li Li [1], Junhuang Xu [2], Yifan Wang [1], Wei Meng [1], Shengping Fan [1] and Hongqiang Li [2,*]

[1] Electric Power Research Institute of Guangdong Power Grid Co., Ltd., Guangzhou 510080, China
[2] School of Materials Science and Engineering, South China University of Technology, Guangzhou 510640, China
* Correspondence: lihq@scut.edu.cn

**Abstract:** In recent years, the number of high-voltage transmission lines has sharply increased with the rapid development of modern industry. However, a corona discharge phenomenon often occurs on the exposed high-voltage transmission lines, leading to energy loss and noise pollution. Herein, we have proposed a facile spraying method to prepare a superhydrophobic and thermally conductive coating to restrain the corona discharge phenomenon of high-voltage transmission lines, with vinyl silicone oil and hydrogen silicone oil as the main materials and modified boron nitride (BN) as a thermal conductive filler. The obtained composite coating exhibited superhydrophobicity, with a high water contact angle of 162°. In addition, the coating also showed a good self-cleaning capability, non-adhesion capability, mechanical stability, and chemical stability. Owing to the construction of the thermally conductive pathways with BN, the thermal conductivity of the coating reached 1.05 W/m·K, which was beneficial to quickly dissipating the heat generated by the current heating effect. Moreover, the corona losses of the positive and negative electrodes under simulated rainy conditions were decreased by 7.43% and 8.05%, respectively. The findings of our work have provided a new strategy to effectively restrain the corona discharge phenomenon of transmission lines, showing great application potential in the field of high-voltage power networks.

**Keywords:** superhydrophobic; thermal conductive; coating; corona discharge; audible noise

## 1. Introduction

Nowadays, with the acceleration of the urbanization process and the rapid development of modern industry, the demand for electric power from first-tier metropolitans and industrial cities has increased obviously, resulting in the continuous climbing of the power load [1–3]. In order to ensure a sufficient power supply, it is necessary to build new substations and erect corresponding high-voltage transmission lines. Generally, high-voltage transmission lines exposed in air are prone to not only suffer local damage or scratches, but to also be polluted by natural salts, dust, and bird droppings [4,5]. Moreover, corona discharge and audible noise will occur under humid or rainy conditions [6–8]. Corona discharge is the ionization phenomenon of the air around the wires under the action of a strong electric field. Due to the uneven electric field generated by the conductor, the electric field intensity around the transmission lines will exceed the critical breakdown field strength of the air (20–30 kV/cm) when the voltage rises to a certain value. The air near the wires will be repeatedly ionized under the action of the electric field to generate sound, light, and heat [9–11]. The above phenomenon not only results in noise pollution into the living environment of the nearby residents, causing misgivings about personal safety, but also leads to power loss [12–16]. Therefore, it is important to suppress the corona discharge of high-voltage transmission lines.

At present, the main methods to restrain the corona discharge in power engineering is to change the structures and properties of the transmission lines or substations, such as by increasing the diameter and number of split wires, raising the height of tower rods, and using spherical parts to hollow out the wires [17–20]. Nevertheless, these methods are often high-cost and difficult to achieve in large-scale applications. In comparison, using a protective coating layer to cover wires has the advantages of ease of operation, low cost, and simple maintenance. However, traditional coatings, such as low-density polyethylene (LDPE) coatings on transmission lines, can only repair the damages and wrinkles on the surface, playing a limited role in improving anti-corona performance [21]. Inspired by natural lotus leaves, water strider legs, and butterfly wings, many researchers have designed and developed superhydrophobic coatings and surfaces with high water contact angles above 150° and water sliding angles of less than 10° [22–26]. Water droplets on these surfaces spontaneously roll off easily and take away contaminants, which is the characteristic of self-cleaning [27,28]. Therefore, the formation of superhydrophobic coatings on transmission lines will not only repair the wrinkles and change the radius of the curvature of the wires, but also weaken the local electric field intensity around the wires. In addition, it can avoid the attachment of water droplets, so as to reduce the number of corona discharge point sources and restrain corona noise. Wu et al. [29] used room-temperature-vulcanized silicone rubber (RTV), liquid silicone rubber (LSR), or permanent-room-temperature-vulcanized composite coatings (PRTV) to coat wires in order to restrain the corona discharge phenomenon on rainy days. However, the corona initiation voltage of the wires coated with an RTV coating was 2.1 times that of the uncoated wires. The water contact angle (WCA) of the above silicone coatings was only about 120°. In addition, the coatings were not thermally conductive and struggled to effectively transfer the heat generated by the current heating effect, which will lead to a potential safety hazard.

In this work, we have proposed a facile method to fabricate a superhydrophobic and thermally conductive composite coating on high-voltage transmission lines to restrain the corona discharge phenomenon. Specifically, we chose vinyl silicone oil and hydrogen silicone oil as the main materials of the coating in order to provide low surface energy. Meanwhile, silica and boron nitride (BN) were used as fillers to construct a rough structure and thermally conductive pathways. The preparation process is illustrated in Figure 1. To improve the dispersion, the silica was modified with fluorosilane, and the BN was treated with sodium hydroxide and then further fluorinated to attach -CF$_3$ groups. Next, the fillers were successively added into the mixture of vinyl silicone oil and hydrogen silicone oil and evenly dispersed under ultrasonication. Finally, the obtained mixture was uniformly sprayed onto the substrate with an airbrush gun and thermally cured through the reaction between V-PDMS and PHMS with Karstedt catalyst to obtain the superhydrophobic and thermally conductive composite coating. The chemical structures of the modified silica and the BN were characterized. The morphology of the composite coating was observed using scanning electron microscopy, and the wettability, adhesion, stability, and thermal conductivity were evaluated. In addition, the suppression effect of the composite coating on the corona discharge of the high-voltage transmission line was also studied.

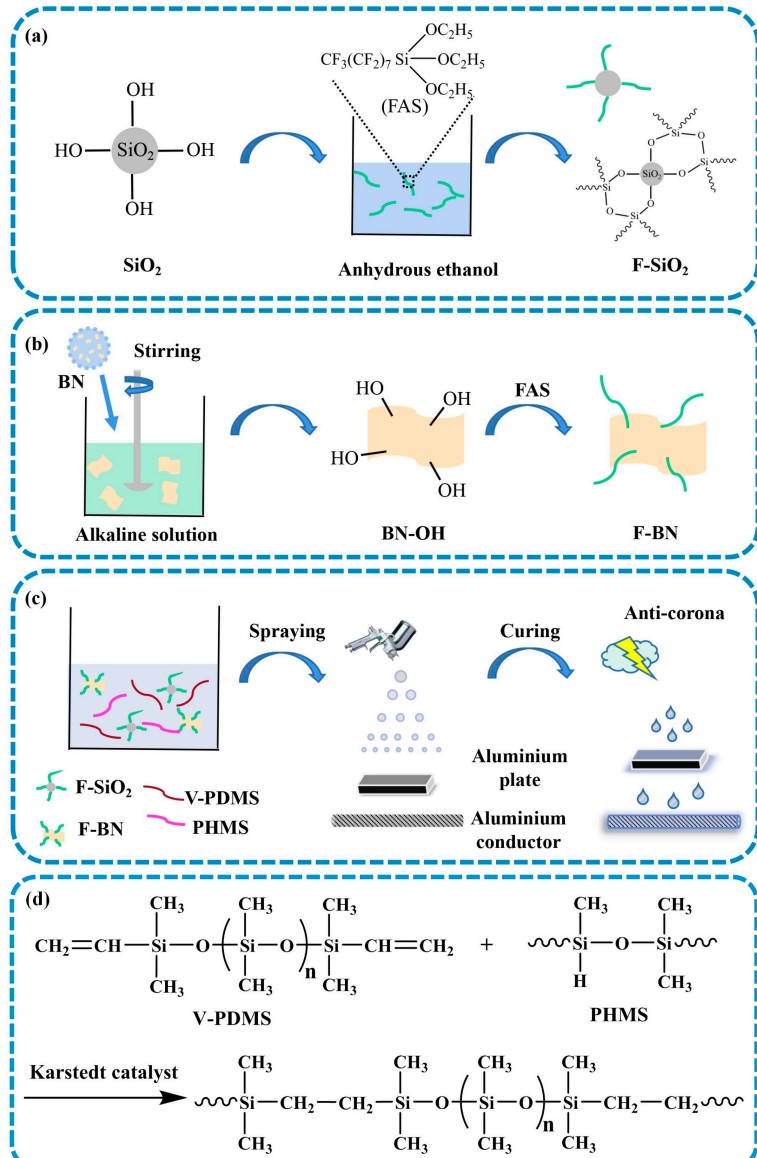

**Figure 1.** (**a**) Schematic diagram of the preparation of fluorinated silica particles. (**b**) Schematic diagram of the modification of BN. (**c**) Schematic illustration of the fabrication of the V-PDMS/F-BN/F-SiO$_2$ composite coating with superhydrophobicity and thermal conductivity for restraining corona loss and audible noise. (**d**) Chemical reaction between V-PDMS and PHMS with Karstedt catalyst.

## 2. Materials and Methods

### 2.1. Materials

The vinyl silicone oil (V-PDMS, with vinyl content of 0.05%–0.22%), hydrogen silicone oil (PHMS, with hydrogen content of 1.5%–1.6%), 1-ethynyl cyclohexanol, Karstedt catalyst (platinum (0)-1,3-divinyl-1,1,3,3-tetramethyldisiloxane, containing 3000 ppm of platinum), and silica particles (SiO$_2$, 30–40 nm) were provided by Guangzhou Siyou New Material Technology Co., Ltd. (Guangzhou, China). The boron nitride powder (BN, 50–60 µm) was supplied by Shanghai Meryer Co., Ltd. (Shanghai, China). The heptafluorodecyl trimethoxysilane (FAS) was purchased from Aladdin Reagent Co., Ltd. (Shanghai, China). The deionized water and n-hexane were obtained from Guangdong Guangshi Reagent Technology Co., Ltd. (Zhaoqiang, China). The aluminum plates (25 mm × 25 mm × 0.28 mm) were purchased from a local store.

### 2.2. Preparation of Fluorinated Silica (F-SiO$_2$) Particles

Firstly, 4 mL of ammonium hydroxide and 6 mL of deionized water were mixed in a beaker. Then, 3 g of silica particles was added into the solution. The above mixture was magnetically stirred for 5 min at room temperature and then treated with an ultrasonic sonicator (FB11201, Fisherbrand, Waltham, MA, USA) for 10 min to obtain a uniform silica suspension. Next, 0.6 mL of 1H,1H,2H,2H-heptafluorodecyl trimethoxysilane was added into 80 mL of anhydrous ethanol and stirred for 10 min. After that, the silica suspension was added into the ethanol solution and stirred continuously for 24 h. The mixture was then centrifuged at 6000 rpm for 10 min, and the precipitate was washed with ethanol and then dried at 50 °C for 3 h, whereafter, the fluorinated silica (F-SiO$_2$) particles were obtained.

### 2.3. Preparation of Fluorinated BN (F-BN) Powders

Firstly, 2 g of the BN powders was added into 30 mL of the sodium hydroxide solution, with a concentration of 2 mol/L, at room temperature. The mixture was fully stirred and heated to 120 °C for 20 h. The product was centrifuged at 2000 rpm for 35 min, washed with deionized water, and then dried in a vacuum oven at 60 °C for 2 h to obtain hydroxylated BN powders (H-BN). Subsequently, 0.5 mL of FAS was added into 80 mL of anhydrous ethanol and constantly stirred for 5 min, and the H-BN powders were added and stirred at room temperature for 6 h. The suspension was centrifuged at 2000 r/min for 35 min, and the precipitate was washed with ethanol and then dried at 60 °C for 2 h, whereafter, the fluorinated BN (F-BN) powders were obtained.

### 2.4. Preparation of V-PDMS/F-BN/F-SiO$_2$ Composite Coating

The aluminum plates were rinsed with anhydrous ethanol and then dried and treated with oxygen plasma at 0.2 mbar for 300 s using a plasma machine (PlasmaFlecto 10, Plasman Technology, Herrenberg, Germany) as substrates for use. Hydrogen silicone oil, 1-ethynyl cyclohexanol as an inhibitor, and Karstedt catalyst were, respectively, diluted with n-hexane to obtain solutions with concentrations of 0.33 wt %, 1 wt %, and 5 wt %, respectively. Next, 1 g of vinyl silicone oil, 0.18 g of 1-ethynyl cyclohexanol solution, 0.75 g of Karstedt catalyst solution, and 0.766 g of hydrogen silicone oil solution were added into 30 mL of n-hexane successively, which was then magnetically stirred for 6 min. Then, 0.2 g of F-BN and 0.4 g F-SiO$_2$ were added and further stirred for 5 min, followed by ultrasonication treatment for 20 min to obtain the dispersion for spraying. Before spraying, the pressure of the spray gun was set at 0.6 MPa, the distance between the spray gun and the substrate was fixed to about 15 cm, and the moving speed was kept at 1 cm/s. The dispersion was uniformly sprayed onto the aluminum plates with forward and reverse moving for one cycle. After curing at 80 °C for 1 h, the superhydrophobic and thermally conductive composite coating was obtained. For comparison, the V-PDMS coating, V-PDMS/F-SiO$_2$ coating, and V-PDMS/F-BN coating were also prepared according to the above procedure.

### 2.5. Evaluation of Adhesion, Mechanical Strength, and Chemical Stability

The adhesion of the V-PDMS/F-BN/F-SiO$_2$ coatings was evaluated according to the standard method of ISO2409-2013. To test the mechanical strength, the V-PDMS/F-BN/F-SiO$_2$ coating layer (25 mm × 20 mm) on the aluminum plate was kept in direct contact with a piece of 400-mesh sandpaper and pulled repeatedly with a load pressure of 50 g, a pulling speed of 2 cm/s, and a pulling distance of 10 cm for one abrasion cycle, and the cycle number was recorded. In addition, to study the damage to the coating caused by water impact, the V-PDMS/F-BN/F-SiO$_2$ coating, placed at a tilt angle of 45°, was continuously scoured with a water stream with a pressure of 100 kPa from a height of 40 cm, and 10 s was recorded as one cycle. To estimate the chemical stability, the V-PDMS/F-BN/F-SiO$_2$ coating on the aluminum plate was, respectively, immersed in 1 M HCl (pH = 1) and 1 M NaOH (pH = 13) solutions for 72 h, and the CA and SA of the coating were measured again.

### 2.6. Measurement of Corona Noise of High-Voltage Transmission Line

The properties of audible noise and corona loss on transmission lines were studied by using a corona cage device composed of a corona cage, grading ring, equalizing sphere, insulator, conductive copper tube, epoxy resin support, and support frame, and the simulated rainfall intensity was set as 7 mm/h during measurement, according to Standard GB/T 28592-2012.

### 2.7. Characterizations

The surface morphologies of samples with different coatings were observed with scanning electron microscopy (SEM, EVO 18, Carl Zeiss Jena, Jena, Germany) at an acceleration voltage of 10 kV. The elemental composition analysis was carried out on an energy dispersion spectroscopy instrument (EDS, X-MaxN 20, Oxford Instruments, Abingdon, Britain), accompanied by SEM. Fourier transform infrared spectroscopy (FTIR) was performed with a Bruker Tensor 27 spectrometer (Bruker Optics, Ettlingen, Germany), from 4000 to 500 $cm^{-1}$, with a resolution of 4 $cm^{-1}$ and scanning times of 16. The water contact angle (WCA) was measured with a contact angle meter (DSA100, Hamburg, Germany) using 5 μL of water droplets as the probe liquid at room temperature. The water sliding angle (WSA) was detected using 10 μL of water droplets with the same contact angle meter. The thermal diffusivity ($\lambda$, $mm^2$/s) was measured with a laser flash thermal conductivity test instrument (LFA467 HyperFlash, Netzsch, Germany), and an infrared (IR) camera (FLIR T450sc) was also utilized to record the temperature changes.

## 3. Results and Discussion

### 3.1. Chemical Characterization of F-SiO₂ and F-BN

Figure 2a shows the FTIR spectra of the $SiO_2$ and the F-$SiO_2$. As can be seen from the FTIR spectrum of the unmodified $SiO_2$, strong absorption peaks at 947 $cm^{-1}$ and 3418 $cm^{-1}$ appeared, which were attributed to the -Si-OH group of native $SiO_2$. The peaks at 1116 $cm^{-1}$ and 800 $cm^{-1}$ were related to the stretching vibration of the Si-O-Si segment [30]. In addition, the absorption peak at 1619 $cm^{-1}$ corresponded to $H_2O$. In the FTIR spectrum of the F-$SiO_2$, the absorption peak at 1199 $cm^{-1}$ was derived from the stretching vibration of the C-F, confirming the successful grafting of fluoroalkyl silane [31]. Figure 2b shows the FTIR spectra of the BN, H-BN, and F-BN. Clearly, the BN, H-BN, and F-BN all had absorption peaks at 814 $cm^{-1}$ and 1372 $cm^{-1}$, which corresponded to the out-of-plane bending and the in-plane stretching vibrations of the BN, respectively. The peak at 3415 $cm^{-1}$ in the spectrum of the H-BN was assigned to the stretching vibration of the -OH group. Moreover, the stretching vibration of the C-F at 1195 $cm^{-1}$ of the F-BN verified the achievement of fluorination modification.

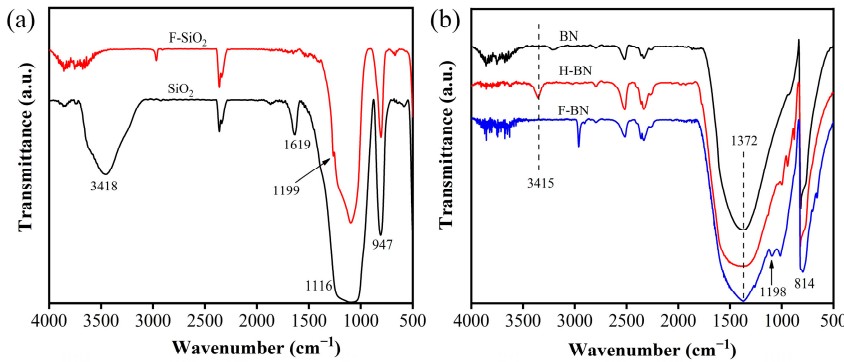

**Figure 2.** (**a**) FTIR spectra of $SiO_2$ and F-$SiO_2$. (**b**) FTIR spectra of BN, H-BN, and F-BN.

### 3.2. Surface Morphology and Wettability of V-PDMS/F-BN/F-SiO₂ Composite Coating

A rough structure is one of the necessary factors required to realize superhydrophobicity. Figure 3 presents the SEM and WCA images of the V-PDMS coating, V-PDMS/F-$SiO_2$

coating, V-PDMS/F-BN coating, and V-PDMS/F-BN/F-SiO$_2$ coating, respectively. As shown in Figure 3a, the V-PDMS coating showed a relatively flat and smooth surface, and the WCA was only 121°. Differently, owing to the introduction of the F-SiO$_2$ nanoparticles, a few protuberances were clearly observed on the surface of the V-PDMS/F-SiO$_2$ coating, and the WCA increased to 153° (Figure 3b), therefore, achieving a superhydrophobic state. In this work, two-dimensional-sheet F-BN was used as a thermally conductive filler in order to endow the coating layer with thermal conductivity. As can be seen in Figure 3c, the V-PDMS/F-BN coating exhibited a rough surface with obvious fluctuations, and the WCA reached 157°. Furthermore, as can be seen from the SEM image of the V-PDMS/F-BN/F-SiO$_2$ coating in Figure 3d, thanks to the introduction of both the F-BN and the F-SiO$_2$, the surface became rougher and showed a multilevel microstructure and the WCA further increased to 162°.

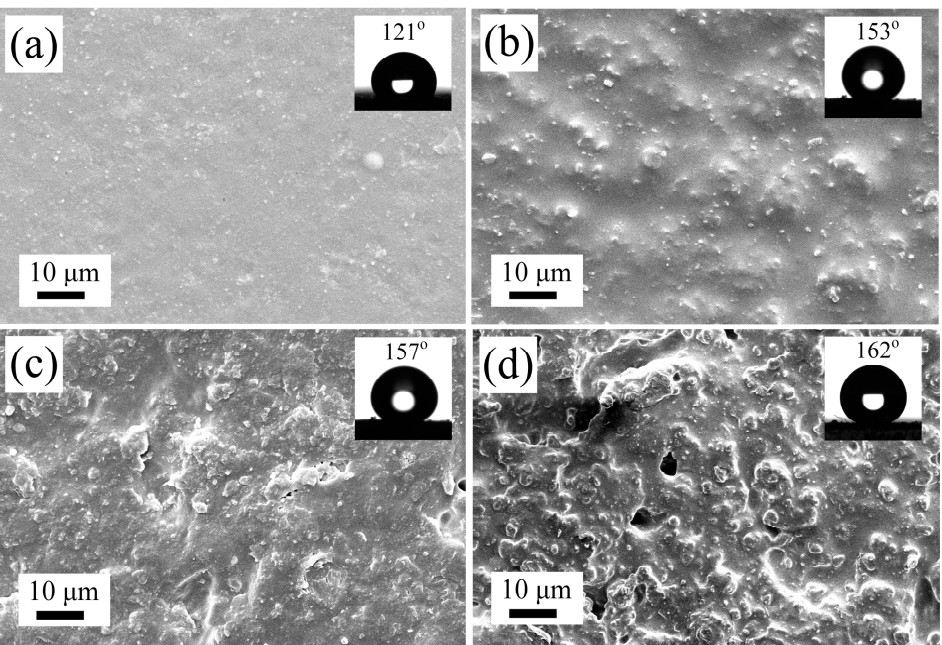

**Figure 3.** SEM images of (**a**) V-PDMS coating, (**b**) V-PDMS/SiO$_2$ coating, (**c**) V-PDMS/F-BN coating, and (**d**) V-PDMS/F-BN/F-SiO$_2$ composite coating.

As shown in Figure 4a, different liquid droplets, including water, milk, and coffee, on the V-PDMS/F-BN/F-SiO$_2$ composite coating were all kept stable, and presented a nearly sphere-like shape. Even after 2 h, the droplets still maintained the same shape, while their size became smaller with the volatilization of the water. Interestingly, when the superhydrophobic V-PDMS/F-BN/F-SiO$_2$ composite coating was immersed into water, a bright, mirror-like surface was observed, as shown in Figure 4b, which was due to the formation of an air layer trapped by the rough structure of the coating surface. The refractive indices of water and air were 1.33 and 1, respectively. Therefore, the total reflection phenomenon occurred when the light traveled from water to the air layer, leading to the appearance of a mirror-like surface [32–34]. In addition, tiny graphite powders were selected as contaminants and were sprinkled onto the coating surface at an inclination angle of 20°, after which the water droplets easily rolled off the surface and carried away the powders (Figure 4c), illustrating the special self-cleaning property of the material. Moreover, when a water droplet was dripped onto the surface of the V-PDMS/F-BN/F-SiO$_2$ composite coating from a height of 4 cm, it quickly bounced and then fell down the surface. Three bounces were observed in this process, fully demonstrating the special non-adhesion property of the composite coating.

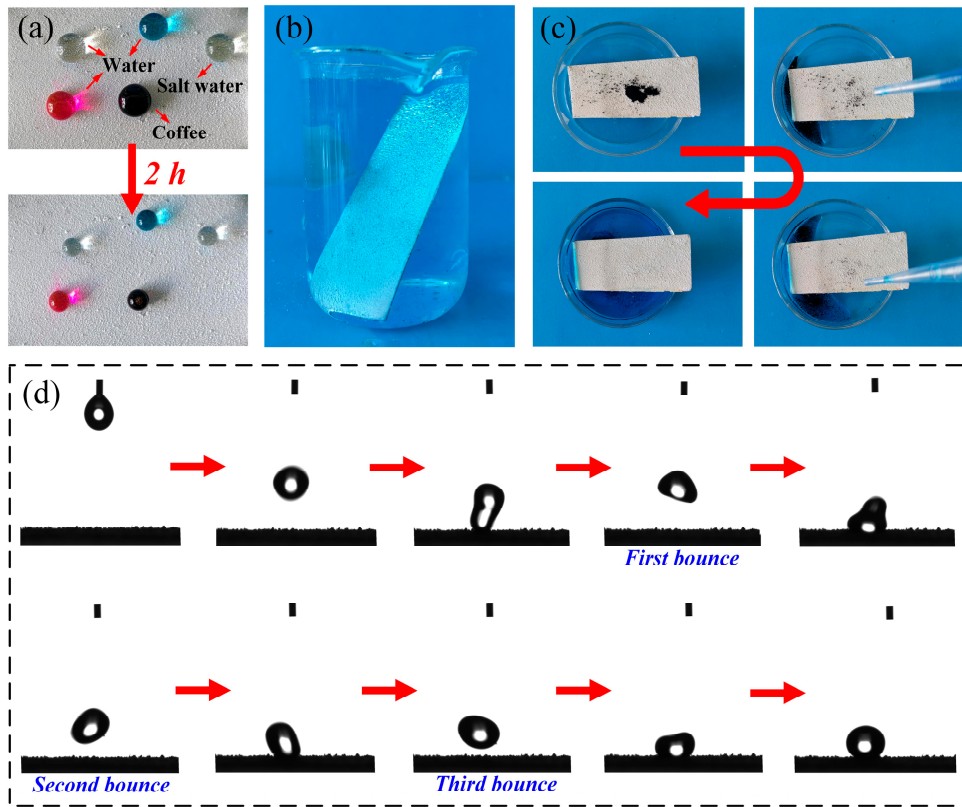

**Figure 4.** (**a**) Photograph of different liquid droplets, including water, salt water, and coffee, on the superhydrophobic coating. (**b**) Photograph of the superhydrophobic coating in water. (**c**) Self-cleaning behavior of the superhydrophobic coating. (**d**) Bounce behaviors of a water droplet on the surface of the superhydrophobic coating.

### 3.3. Adhesion, Mechanical Stability, and Chemical Stability of V-PDMS/F-BN/F-SiO$_2$ Composite Coating

In long-term practical application, the strong adhesion of the V-PDMS/F-BN/F-SiO$_2$ composite coating is necessary. In this work, the adhesion of the V-PDMS/F-BN/F-SiO$_2$ composite coating on the aluminum plate was measured according to Standard ISO2409-2013, no coating layer was removed with tape, and the adhesion reached 0 grade. The high adhesion was possibly because the hydrogen silicone oil and the vinyl silicone oil easily permeated into the tiny defects, such as cracks and micropores, and then formed a mechanical interlock after curing. In order to evaluate the mechanical stability of the V-PDMS/F-BN/F-SiO$_2$ composite coating, abrasion resistance and water shock resistance tests were performed, and the schematic illustrations and digital photos of such are presented in Figure 5a. As shown in Figure 5b, with the abrasion cycle increasing, the WCA of the V-PDMS/F-BN/F-SiO$_2$ coating slightly decreased, yet still remained above 150°, even after 20 abrasion cycles. As can be seen in Figure 5c, the WCA of the coating was almost unaffected by the water stream, and only decreased from 157° to 154° after 50 cycles. Furthermore, the V-PDMS/F-BN/F-SiO$_2$ composite coating was also immersed into different solutions in order to evaluate the chemical stability. The results showed that the WCA of the coating still remained above 150° after immersion in water, acetone, ethanol, toluene, and hexane for 96 h and HCl solution of pH 1 and NaOH solution of pH 13 for 72 h (Figure 5d), indicating an excellent chemical stability. The excellent mechanical stability and chemical stability of the composite coating were mainly due to the formation of an organosilicon crosslinking network.

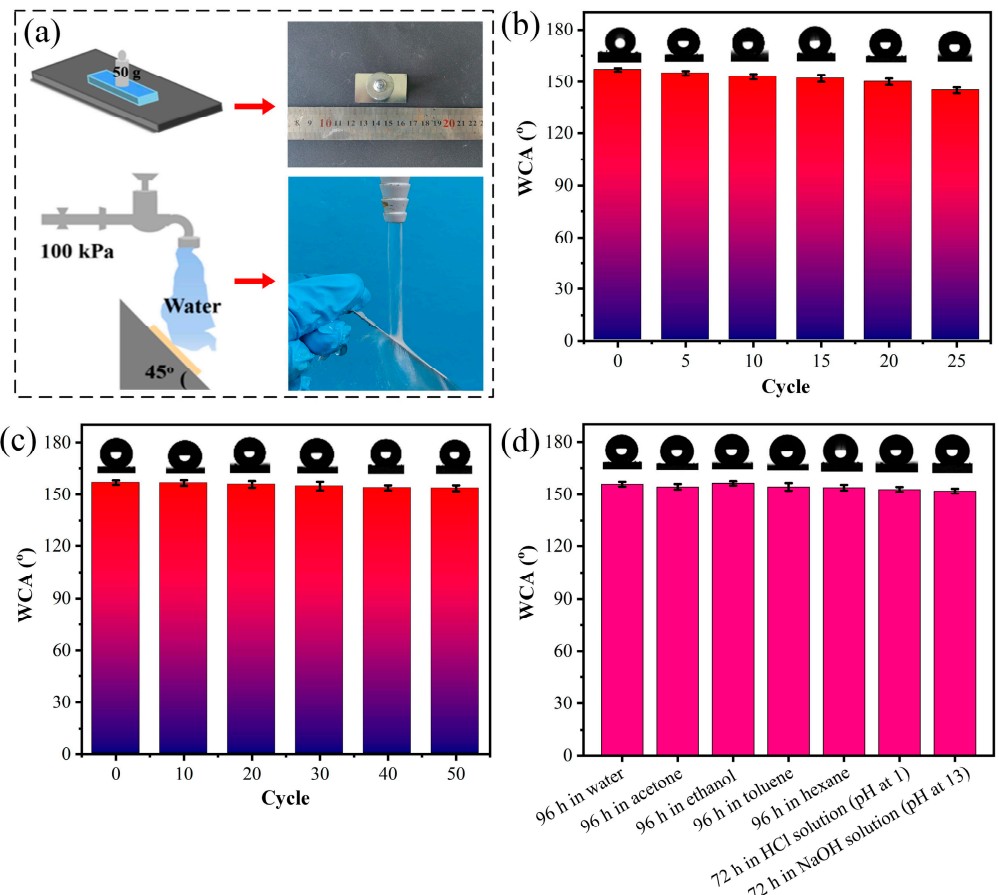

**Figure 5.** (**a**) Schematic illustrations and digital photos of abrasion resistance test and water shock resistance test, respectively. (**b**) WCA of the V-PDMS/F-BN/F-SiO$_2$ composite coating after different abrasion cycles. (**c**) WCA of the V-PDMS/F-BN/F-SiO$_2$ composite coating after different water shock cycles. (**d**) WCA of the V-PDMS/F-BN/F-SiO$_2$ composite coating after immersion in different solutions.

### 3.4. Thermal Conductivity of V-PDMS/F-BN/F-SiO$_2$ Composite Coating

In general, the thermal conductivity of the coating layer on the surface of a high-voltage transmission line is important in practical applications. If the thermal conductivity is relatively poor, the heat generated by the current heating effect of the high-voltage transmission lines will be unable to quickly dissipate, probably leading to local overheating, and even an accident. Owing to the construction of thermally conductive pathways with BN, the thermal conductivity of the V-PDMS/F-BN/F-SiO$_2$ composite coating reached 1.05 W/m·K, which was beneficial for the quick dissipation of the heat generated by the current heating effect. In addition, an incandescent lamp (300 W) was used to irradiate the back side of a piece of aluminum plate coated with V-PDMS/F-BN/F-SiO$_2$ coating at a distance of 10 cm, and the surface temperature was monitored with an infrared thermal imager in real time. As shown in Figure 6, when the irradiation time was 2 s, the surface temperature of the V-PDMS/F-BN/F-SiO$_2$ composite coating increased from 27 °C to 61.3 °C. When the irradiation time was prolonged to 4 s, 6 s, and 8 s, the surface temperature further increased to 71.2 °C, 87.5 °C, and 94.2 °C, respectively, demonstrating an excellent thermal conduction capability. In comparison, the increasing speed of the surface temperature of the V-PDMS/F-SiO$_2$ composite coating on the aluminum plate was relatively slower. The surface temperature increased from 27 °C to 41.2 °C, 58.7 °C, 72.1 °C, and 83.4 °C after irradiating for 2 s, 4 s, 6 s, and 8 s, respectively, indicating the importance of BN for the improvement of the thermal conductivity of the composite coating.

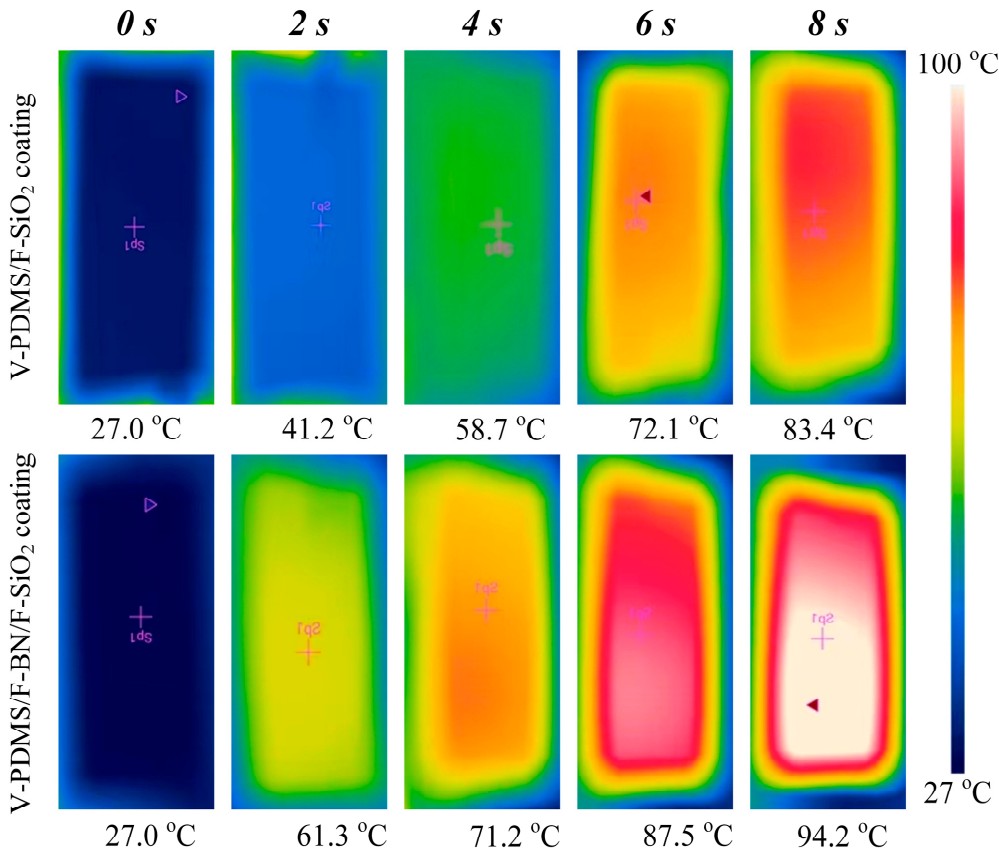

**Figure 6.** Thermal infrared images and surface temperatures of the V-PDMS/F-SiO$_2$ coating and V-PDMS/F-BN/F-SiO$_2$ coating during the heating process.

*3.5. Corona Characteristics of V-PDMS/F-BN/F-SiO$_2$ Composite Coating on High-Voltage Transmission Line*

The audible noise on the high-voltage transmission line was studied using a corona cage device, as illustrated in Figure 7a. In this work, three different pieces of aluminum high-voltage transmission line were selected for measurement and comparison, including blank aluminum wire with a WCA of 78°, V-PDMS-coated aluminum wire with a WCA of 121°, and V-PDMS/BN/SiO$_2$-coated aluminum wire with a WCA of 162°. The sound pressure level and the A-weighting sound pressure level (A-WSPL) of the positive and negative electrodes of these wires were measured under simulated rainy conditions. As shown in Figure 7b, the sound pressure level and the A-WSPL of the coated wires were obviously reduced under the simulated rainy conditions, in comparison with the blank wire, and those of the V-PDMS/F-BN/F-SiO$_2$-coated aluminum wire were the lowest. In addition, the corona loss of the positive electrode coated with the superhydrophobic V-PDMS/F-BN/F-SiO$_2$ composite coating was reduced by 7.43%, and that of the negative electrode was reduced by 8.05%. This was mainly attributed to the fact that the water droplets on the coating surface quickly slipped off, thus leading to a decrease in the number of corona points. Therefore, the superhydrophobic characteristics played a prominent role in the suppression of the corona discharge of the high-voltage transmission line.

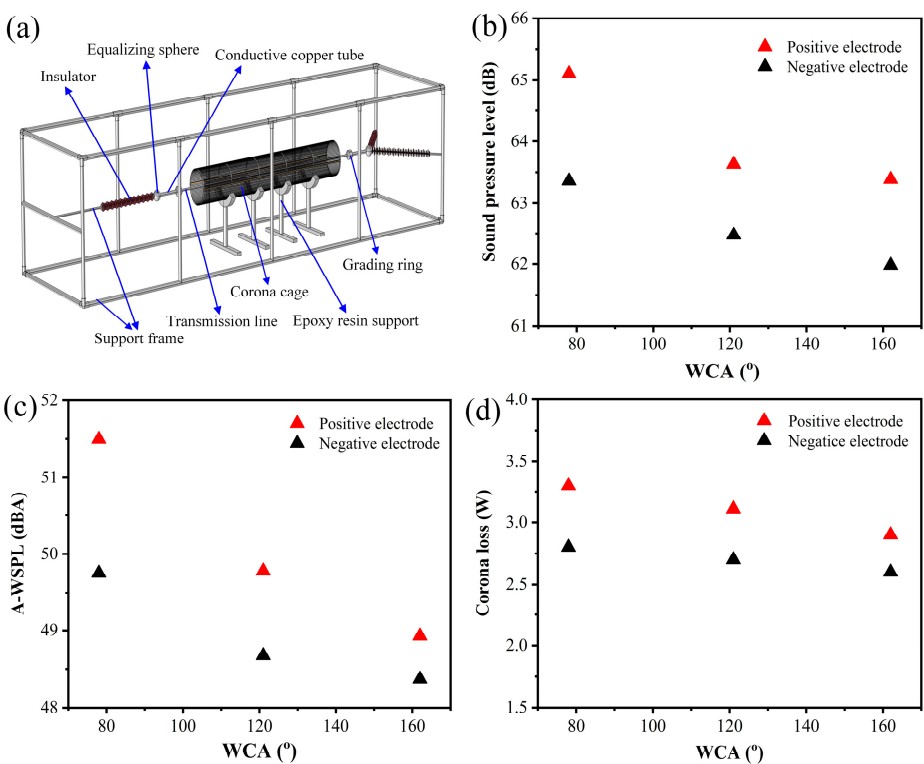

**Figure 7.** (**a**) Schematic illustration of the corona cage device. (**b**) Sound pressure level, (**c**) A-weighting sound pressure level (A-WSPL), and (**d**) corona loss of the positive and negative electrodes of the aluminum high-voltage transmission line coated with V-PDMS/F-BN/F-SiO$_2$ composite coating under simulated rainy conditions, respectively.

## 4. Conclusions

In summary, we successfully prepared a superhydrophobic and thermally conductive V-PDMS/F-BN/F-SiO$_2$ composite coating for restraining the corona loss and audible noise of a high-voltage transmission line. The coating showed superhydrophobicity, with a water contact angle of 162°, and appeared to have excellent water non-adhesion characteristics and self-cleaning capabilities. In addition, the coating had a strong adhesion force of grade 0 on the aluminum substrate and favorable mechanical stability and chemical stability. The thermal diffusion coefficient of the coating reached 1.05 W/m·K, which was conducive to reducing the energy dissipation during the operation of the transmission lines. Compared with the blank wire, the sound pressure level and the A-WSPL of the coated wires were obviously reduced under simulated rainy conditions. The corona loss of the positive electrode and the negative electrode decreased by 7.43% and 8.05%, respectively. The prepared superhydrophobic and thermally conductive V-PDMS/F-BN/F-SiO$_2$ composite coating has great potential in the field of high-voltage and super-high-voltage transmission lines for the suppression of the corona discharge phenomenon.

**Author Contributions:** Conceptualization, L.L. and J.X.; methodology, J.X.; software, Y.W. and W.M.; validation, L.L. and S.F.; formal analysis, J.X. and H.L.; investigation, L.L. and J.X.; resources, Y.W. and W.M.; data curation, W.M. and S.F.; writing—original draft preparation, J.X.; writing—review and editing, H.L.; visualization, J.X.; supervision, H.L.; project administration, Y.W.; funding acquisition, L.L. All authors have read and agreed to the published version of the manuscript.

**Funding:** This research was funded by China Southern Power Grid Co., Ltd., grant number GD-KJXM20201968.

**Institutional Review Board Statement:** Not applicable.

**Informed Consent Statement:** Not applicable.

**Data Availability Statement:** Not applicable.

**Conflicts of Interest:** The authors declare no conflict of interest.

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
