# Peer review of "Superhydrophobic and Thermally Conductive Coating for Restraining Corona Loss and Audible Noise of High-Voltage Transmission Lines"

_coatings, doi:10.3390/coatings13091530_

Round 1

Reviewer 1 Report

The article is devoted to a very interesting and topical topic of creating superhydrophobic coatings for protecting power lines. However, despite the work done by the authors, the draft contains serious shortcomings in terms of scientific novelty, test methods and analysis of the results; in this edition, the draft cannot be recommended for publication.

1) Abstract. "surface stability" - what do you mean?

2) The last paragraph of the introduction. Why are there conclusions? There is a special section for conclusions. Show the idea of the work, scientific novelty, your backlog in this area (if any) and the main points that you will explore and discuss in the article.

3) The choice of silicone oils is completely incomprehensible (you write to reduce SFE, but also use FAS to modify particles, and FAS has even less SFE).

4) Platinum catalyst - what is it? Why is it needed? Give its chemical formula. How did you regenerate it?

5) Are your silicone oils crosslinked by heat treatment? Give the chemical formulas of the reagents and the chemical reactions of the curing process.

6) From the SEM photo it can be seen that the modified surfaces are almost smooth, no hierarchical structure is observed.

7) Clauses 3.3, 3.4, 3.5. Provide a detailed description of the test methods in a separate section.

8) Clause 3.4. It is necessary to compare with pure (original) aluminum. Maybe your coatings degrade the properties of aluminum.

9) There is no analysis of the results in the draft - only a statement: "we got such and such a result." It is necessary to explain why such results are obtained; how coating components, surface microstructure, etc. influence.

10) What wire was used in the corona test? How was the wire coated?

Reviewer 2 Report

The manuscript entitled "Superhydrophobic and Thermally Conductive Coating for Restraining Corona Loss and Audible Noise of High-Voltage Transmission Lines” reports a facile fabrication method of a superhydrophobic and thermally conductive composite coating for a high voltage transmission line. The authors demonstrated in detail that the special wettability and superior heat conductivity can work synergistically to restrain corona loss and audible noise. The work is quite detailed and would be of interest to the scientific community. In my point of view, the manuscript is recommended for publication in Coatings after minor revision. Detailed comment is shown below.

1)    It is necessary to demonstrate that the surface structure, wettability, mechanical stability, adhesion, and other properties of the coating layer can be maintained under prolonged thermal exposure. The durability of the coating against heat generated by the current heating effect is very important for the practical application on high-voltage transmission lines.

2)    It seems that "Figure 9" in Line 259 is a typo. Please correct it to "Figure 6".

Author Response

Reviewer 2

Overall Comments: The manuscript entitled "Superhydrophobic and Thermally Conductive Coating for Restraining Corona Loss and Audible Noise of High-Voltage Transmission Lines” reports a facile fabrication method of a superhydrophobic and thermally conductive composite coating for a high voltage transmission line. The authors demonstrated in detail that the special wettability and superior heat conductivity can work synergistically to restrain corona loss and audible noise. The work is quite detailed and would be of interest to the scientific community. In my point of view, the manuscript is recommended for publication in Coatings after minor revision.

Response: We thank the reviewer for reading through our manuscript and providing positive comment.

Comment 1: It is necessary to demonstrate that the surface structure, wettability, mechanical stability, adhesion, and other properties of the coating layer can be maintained under prolonged thermal exposure. The durability of the coating against heat generated by the current heating effect is very important for the practical application on high-voltage transmission lines.

ResponseWe agree with the reviewer that the durability of the coating against heat generated by the current heating effect is very important for the practical application on high-voltage transmission lines. So, we used crosslinked organosilicon as the matrix of the composite coating in this work. As well known, due to the strong Si-O bonds, organosilicon has outstanding heat resistance and can be used at about 250 oC for long term. In general, the surface temperature of the high-voltage transmission lines caused by current heating effect is lower than 90 oC and even 60 oC in practical application. Therefore, we think that the properties of the composite coating can be maintained under prolonged thermal exposure.

Comment 2: It seems that "Figure 9" in Line 259 is a typo. Please correct it to "Figure 6".

Response: Thanks. We have corrected it in the manuscript.